# Effect of Asphalt Pavement Base Layers on Transverse Shrinkage Cracking Characteristics

Mingming Xing [1], Hongwei Yang [2,*], Zhenguo Zhao [1] and Tengjiang Yu [3,*]

[1] Heilongjiang Provincial Communications Planning and Design Institute Group Co., Ltd., Harbin 150080, China
[2] Heilongjiang Bada Highway Engineering Co., Ltd., Harbin 150028, China
[3] School of Architecture Engineering, Weifang University, Weifang 261061, China
* Correspondence: mxgsyhw@163.com (H.Y.); tengjiang1025@foxmail.com (T.Y.)

**Abstract:** Transverse shrinkage cracking is considered to be one of the main factors leading to the degradation of sustainable service performance for asphalt pavement, but the effect of base layers on it is rarely studied. To investigate the effect of asphalt pavement base layers on transverse shrinkage cracking characteristics, this study analyzed and evaluated it through the test section data. The transverse shrinkage cracking test section data of four different asphalt pavement base layers were compared, and the variation characteristics of transverse shrinkage cracking under the influence of the base layer types and temperature were analyzed. Based on the conventional characteristics, the concept and calculation method of cracking area ratio (CAR) was proposed, and its rationality and accuracy were proved by calculation. It has been found that the CAR can reflect the longitudinal shrinkage performance of asphalt pavement. The shrinkage rate of a semi-rigid base is larger, while the shrinkage performance of a flexible base is better, and the shrinkage performance of the whole pavement is not affected by a small amount of cement. The research provides a practical basis for improving the anti-cracking performance of asphalt pavement and helps to enhance its sustainable service performance.

**Keywords:** asphalt pavement; base layers; transverse shrinkage cracking; cracking area ratio; sustainable service performance

## 1. Introduction

Transverse shrinkage cracking is considered to be one of the most common problems of asphalt pavement, especially in cold environments [1]. The pavement interior is affected due to precipitation through the transverse shrinkage cracking, resulting in the asphalt pavement structure being damaged by frost swelling [2,3]. The durability of asphalt pavement and the driving experience of vehicles are reduced, which is not conducive to the sustainable service performance of asphalt pavement. Relevant studies show that the transverse shrinkage cracking of asphalt pavement is closely related to the performance of the pavement structural layer, especially the material of base layers [4–6]. Therefore, it is an important research direction to study the effect of asphalt pavement base layers on transverse shrinkage cracking characteristics, which is helpful to increase the durability of the asphalt pavement structural layer in cold areas.

At present, the research on reducing the transverse shrinkage cracking of asphalt pavement in cold areas mainly focuses on improving the performance of base layer materials [7,8]. For example, Sha [9] has explored the damage model of a semi-rigid base through many experimental studies to describe and predict the main causes of water damage performance for semi-rigid materials. Tian [10] found that the tensile stress of the lower base was greater than that of the upper base, and the fatigue life of the pavement structural layer depended on the base layer characteristics. Qi [11] studied the dynamic and static spring back modulus tests and field CBR tests of continuously graded crushed

stone base materials in the laboratory and found that with an increase in the load capacity, the modulus of graded crushed stone materials gradually increased. Li [12] compared the mechanical characteristics of semi-rigid pavement and flexible pavement and obtained the influence law of different structural layer thicknesses on pavement load and the mechanical response law and trend of asphalt pavement structural layer. Through a lot of research on the mechanical properties of base layer material, the load capacity of base layer material was constantly optimized and improved, which was conducive to reducing transverse shrinkage cracking of asphalt pavement [13,14]. However, there are differences between laboratory experimental research and engineering service environmental conditions, and the laboratory test indexes of base layer material have certain limitations.

Considering the difference between engineering service environmental conditions and laboratory tests, it has become the focus of many researchers to study the performance of asphalt pavement base layer material from statistical data in engineering projects [15,16]. For example, Xin [17] studied the influence of vehicle speed and traffic volume on its dynamic response and service life through practical engineering projects, and field dynamic strain measurements indicated that a low speed amplified the impact caused by the wheel load effect. Cui [18] investigated the application state of a flexible base under asphalt pavement of Chengdu-Chongqing high-speed and analyzed some problems affecting the subordinate gravel (flexible base) in traffic congestion. Therefore, data from engineering projects can enrich the study of asphalt pavement base layers, which can lead to improving the base layer performance [19,20].

In conclusion, numerous laboratory studies and engineering projects have achieved positive results in exploring the effect of base layer material on the asphalt pavement structural layer. However, the effect of asphalt pavement base layers on transverse shrinkage cracking characteristics has not been studied and explored. Therefore, based on the asphalt pavement test section, this study summarizes the parameters of transverse shrinkage cracking characteristics for asphalt pavement induced by different base layers through actual investigation, and the proposed cracking area ratio (*CAR*) could reasonably describe the characteristics of transverse shrinkage cracking.

## 2. Project and Survey Results

### 2.1. Test Section Project Information

The asphalt pavement test section is located in the Songnen Plain of Heilongjiang Province in the middle temperate zone of the Northern Hemisphere [21]. The geographical location ranges from 124°13′ to 128°30′ east longitude and 45°3′ to 48°02′ north latitude. Additionally, the climate of the project was characterized by the same period of rain and heat, winter covered by snow and ice, and concentrated summer rainfall. The average annual temperature is between 1.3 °C and 4.0 °C, the frost-free period is from 120 to 140 days, the sunshine duration is from 2600 to 2900 h, and the average precipitation is 483 mm [22]. The geographic location of the project is shown in Figure 1.

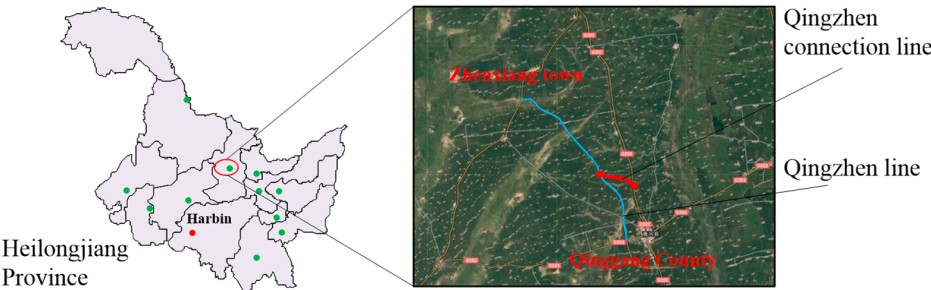

**Figure 1.** Geographic location of the project.

The asphalt pavement test section is located at K2+204~K2+846 of Qingzhen connection line, and includes the 225 m graded gravel base section (flexible base) of the test section K2+425~K2+650 [23]. Meanwhile, the conventional 5.5% cement stable base section

(semi-rigid base) has a length of 200 m on each side of the test section. The structural layer thickness and material of the asphalt pavement test section are shown in Figure 2.

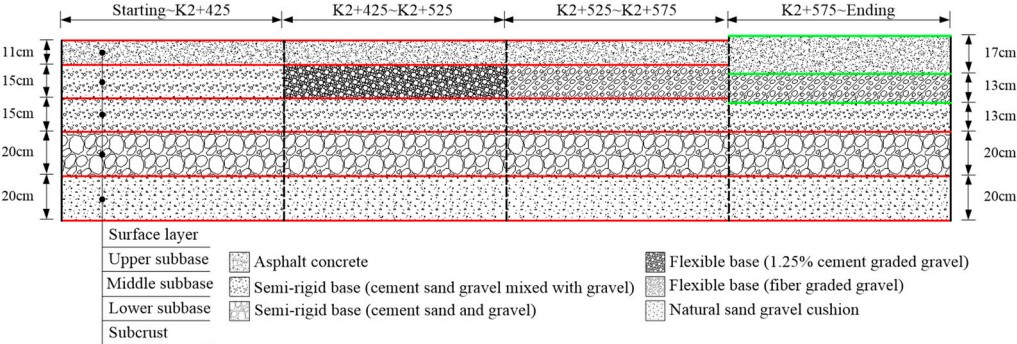

**Figure 2.** Asphalt pavement test section structural layer parameters.

In the investigation of transverse shrinkage cracking, it can be found that there are four transverse cracks at K2+421, K2+520, K2+572, and K2+650, respectively. Combined with the structural layer information of the asphalt pavement test section (as shown in Figure 2), it can be seen that the transverse cracks in K2+421, K2+520, K2+572, and K2+650 are mainly caused by the complex stress of pavement structural layer materials [24,25]. Therefore, in the calculation and analysis, the problems of section division have been fully considered.

*2.2. Survey Results*

The cracking investigation of the asphalt pavement test section is mainly based on the current Chinese evaluation standards and design specifications, including Highway Performance Assessment Standards (JTG 5210-2018) and the latest version of Specifications for Design of Highway Asphalt Pavement (JTG D50-2017) [26,27]. However, the project is located in a heavy freezing area, and the cracks of newly built asphalt pavement are mainly transverse shrinkage cracking. Therefore, the statistics were mainly focused on the transverse shrinkage cracking, which is convenient to analyze the influence of base layer material.

In the investigation, a tape measure was used to calibrate the cracking pile number, and a steel tape measure was used to measure the cracking width (0.1 mm). Additionally, the investigation contents of transverse shrinkage cracking include cracking distance, width, and length. The survey was carried out three times as follows: on 6 April 2018 (initial thaw period); on 21 October 2018 (initial freezing period); and on 21 March 2019 (late freezing period). The transverse shrinkage cracking of asphalt pavement test section is shown in Figure 3, and the investigation results are shown in Table 1.

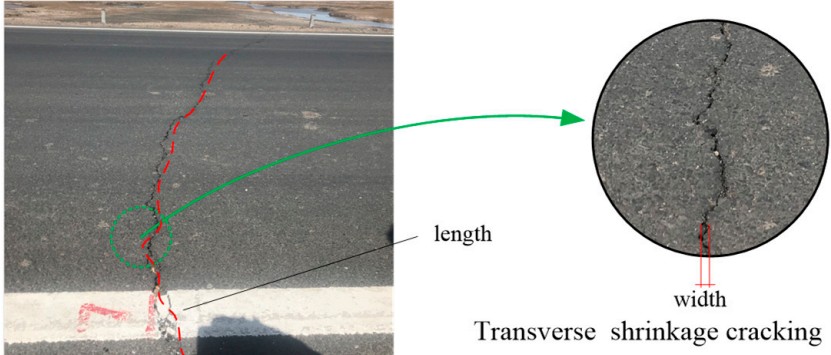

**Figure 3.** Transverse shrinkage cracking of asphalt pavement.

Through the survey data (as shown in Table 1), we can directly obtain the number of transverse cracks and the total width of asphalt pavement. However, it is difficult to

describe and evaluate the transverse shrinkage cracking based on these two parameters because of the difference in length. Therefore, the project needs to conduct an in-depth analysis of the characteristics of transverse shrinkage cracking in asphalt pavement in order to describe the relationship more reasonably between transverse shrinkage cracking and its base layers.

**Table 1.** Survey results of transverse shrinkage cracking for asphalt pavement.

| Date | Test Section | Section Length $L$/(m) | Number of Transverse Cracks | Total Width $Bs$/(mm) |
|---|---|---|---|---|
| 6 April 2018 | Starting~K2+425 | 188 | 8 | 30.8 |
| | K2+425~K2+525 | 105 | 4 | 10.5 |
| | K2+525~K2+575 | 122 | 3 | 12.3 |
| | K2+575~Ending | 126 | 6 | 22.5 |
| 21 October 2018 | Starting~K2+425 | 241 | 11 | 33.3 |
| | K2+425~K2+525 | 105 | 4 | 7.3 |
| | K2+525~K2+575 | 122 | 3 | 4.8 |
| | K2+575~Ending | 196 | 8 | 28.0 |
| 21 March 2019 | Starting~K2+425 | 241 | 12 | 62.5 |
| | K2+425~K2+525 | 105 | 5 | 17.5 |
| | K2+525~K2+575 | 122 | 3 | 16.3 |
| | K2+575~Ending | 196 | 9 | 42.3 |

## 3. Analysis of Transverse Shrinkage Cracking Characteristics in Asphalt Pavement

### 3.1. Characteristics of Transverse Shrinkage Cracking

The number of transverse cracks and the total width are not enough to explain the influence of the asphalt pavement base layer on transverse shrinkage cracking [28–30]. Therefore, the transverse shrinkage cracking was analyzed by calculating the mean cracking distance and averaged cracking width. Meanwhile, in order to describe the characteristics of transverse shrinkage cracking of asphalt pavement more accurately, we have defined a new technical term (parameter), namely, the cracking area ratio (*CAR*), by combining the mean cracking distance and averaged cracking width. The calculation formula is shown in Formula (1).

$$CAR = \frac{Bs \times B}{L \times B} = \frac{Bs}{L} = \frac{Bs}{10 \times L} \tag{1}$$

where, *CAR* is the cracking area ratio, (cm/100 m); *Bs* is the total width of cracks in the test section, (mm); *L* is the test section length, (m); *B* is the pavement width.

According to Formula (1), *CAR* is the proportion of cracking area in the total road surface area of the whole section, which is numerically equivalent to "the ratio of total cracking width (longitudinal length along the road) to total road surface length of the whole section". Therefore, for the transverse shrinkage cracking of asphalt pavement, *CAR* has the following meanings:

(1) It can evaluate the overall situation of transverse shrinkage cracking distribution in asphalt pavement more intuitively, accurately, and comprehensively.
(2) It can directly judge the overall shrinkage state of the road surface and the rainwater infiltration amount on the road surface.
(3) The method of calculating *CAR* directly without classification can make up for the weakness or defect of "single boundary and general and rough boundary".

Therefore, compared with the two intuitive parameters of mean cracking distance and averaged cracking width, *CAR* can comprehensively analyze the transverse shrinkage cracking detection data, and has the advantages of being intuitive, comprehensive, scientific, reasonable, and practical. Based on Table 1, the mean cracking distance, averaged cracking width, and *CAR* were calculated, and the calculation results are shown in Table 2.

**Table 2.** Calculation results of transverse shrinkage cracking of asphalt pavement.

| Date | Base Layer | Base Layer I | Base Layer II | Base Layer III | Base Layer IV |
|---|---|---|---|---|---|
| 6 April 2018 | Mean cracking distance/(m) | 23.4 | 24.8 | 43.3 | 21.3 |
| | Averaged cracking width/(mm) | 3.84 | 2.63 | 4.08 | 3.75 |
| | *CAR* (cm/100 m) | 1.64 | 1.01 | 1.00 | 1.79 |
| 21 October 2018 | Mean cracking distance/(m) | 21.7 | 23.1 | 43.1 | 24.5 |
| | Averaged cracking width/(mm) | 3.02 | 1.81 | 1.58 | 3.50 |
| | *CAR* (cm/100 m) | 1.38 | 0.69 | 0.39 | 1.43 |
| 21 March 2019 | Mean cracking distance/(m) | 19.7 | 19.8 | 43.2 | 21.8 |
| | Averaged cracking width/(mm) | 5.21 | 3.50 | 5.42 | 4.69 |
| | *CAR* (cm/100 m) | 2.59 | 1.67 | 1.33 | 2.16 |

Note: Base layer I represents asphalt pavement with cement sand gravel mixed with gravel as the upper base. Base layer II represents asphalt pavement with 1.25% cement graded gravel as the upper base. Base layer III represents asphalt pavement with fiber graded gravel (two surface layers) as the upper base. Base layer IV represents asphalt pavement with fiber graded gravel (three surface layers) as the upper base.

It can be seen from Table 2 that the mean cracking distance of the fiber-graded gravel base was much larger than that of other base layers, but the average cracking width was also slightly larger than that of other base layers. Although the performance of transverse shrinkage cracking in fiber-graded gravel base was relatively better than the multiple relations, it was not easy to compare the advantages and disadvantages of each situation using the two parameters of mean cracking distance and averaged cracking width. According to the *CAR*, due to the small mean cracking distance and large averaged cracking width, the *CAR* of the Base layer I test section was 1.64 cm/100 m. For the Base layer II test section, the mean cracking distance was not much improved, and the averaged cracking width was significantly reduced, so the *CAR* was calculated as 1.01 cm/100 m. On the other hand, in the Base layer III test section, the mean cracking distance was large. Additionally, the averaged cracking width was also the largest 4.08 mm in this set of data, and the *CAR* was calculated as 1.00 cm/100 m.

### 3.2. Effect of Base Layer Types on Transverse Shrinkage Cracking

The mean cracking distance and averaged cracking width are related to the thickness and material properties of asphalt concrete and base layer, as well as the bonding between layers, and the material properties of the base layer play an important role. When the surface layer of asphalt pavement is the same, the influence factors of transverse shrinkage cracking are mainly affected by the base layer types [31,32]. This is because the asphalt surface layer is directly in contact with the base layer and the process of shrinkage is restricted by the base layer, and thus stress accumulation occurs.

In the project, the base layer material was mainly divided into a semi-rigid base and a flexible base, among which the flexible base was divided into cement and fiber doped. The calculation results of mean cracking distance, average cracking width, and *CAR* for transverse shrinkage cracking of asphalt pavement with different base layer types are shown in Figure 4.

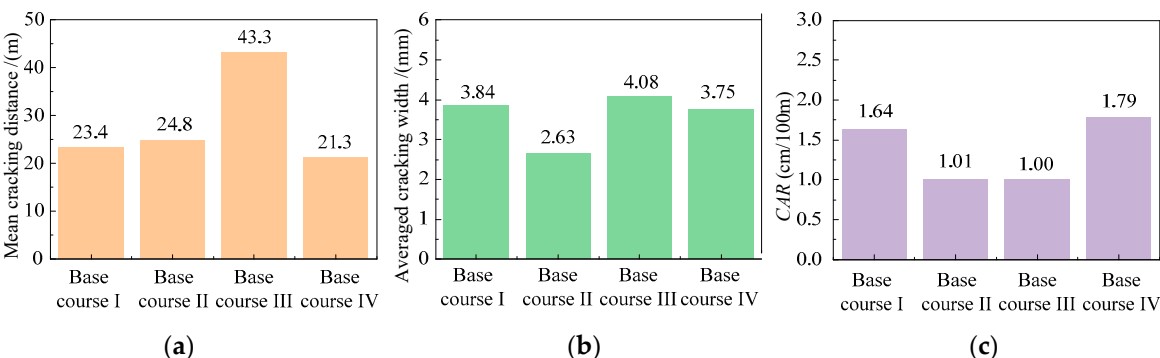

**Figure 4.** Parameters of transverse shrinkage cracking. (**a**) Mean cracking distance; (**b**) averaged cracking width; (**c**) *CAR*.

It can be found from the mean cracking distance (as shown in Figure 4a) that the average of the Base layer I test section was 23.4 m, which belongs to the normal value in Heilongjiang Province. The Base layer III test section was outstanding, reaching 43.3 m with a maximum distance of about 50.0 m. Additionally, the Base layer II test section mixed with 1.25% cement was 24.8 m, slightly higher than the Base layer I test section. It can be seen that a pure loose granular base can make the pavement mean cracking distance increase by about twice, but it will be close to the level of the Base layer I test section as long as a small amount of cement is mixed.

From the averaged cracking width (as shown in Figure 4b), the average of the Base layer I test section was 3.80 mm and the two sections were more uniform. The test section of the fiber graded gravel base mixed with 1.25% cement was 2.63 mm, 30% less than the Base layer I test section. In addition, the test section of fiber graded gravel base was the widest, reaching 4.08 mm, and the largest cracking width of 10.5 mm and the smallest cracking width of 0.5 mm appear in the whole test section, showing obvious non-uniformity.

According to the *CAR* (as shown in Figure 4c), the average Base layer I test section was 1.72 cm/100 m. The fiber graded gravel (with or without 1.25% cement) base layer showed a significant advantage, with a 40% reduction of 1.00 cm/100 m compared to the conventional cement sand gravel mixed with gravel.

In other words, for fiber graded gravel, the mean cracking distance and the averaged cracking width of transverse shrinkage cracking are different with or without mixing 1.25% cement, but the *CAR* is similar. It can be seen that the Base layer II test section was used in the formation period of transverse shrinkage cracking. Due to its strength (small) and dry shrinkage characteristics, it exhibits similar properties (small mean cracking distance) to the Base layer I test section. However, after cracking formation in the continuous shrinkage period (cracking width development period), the fiber graded gravel base mixed with a small amount of cement quickly and evenly loses strength (strength failure) under the action of additional stresses such as driving load. It shows a similar function to a simple loose Base layer III test section. Therefore, the load stress of the asphalt concrete surface can be effectively alleviated and the upward reflection of the cracking in the middle and lower base can be effectively prevented so that the overall shrinkage, and the small, averaged cracking width and *CAR* are the same as that of the simple loose fiber graded gravel base.

## 4. Effect of Temperature on Transverse Shrinkage Cracking

### 4.1. Effect of Temperature on Transverse Shrinkage Cracking Evaluation Indexes

The asphalt pavement test section is located in the heavy freezing area of Heilongjiang Province, and there is a temperature difference of −35 to −60 °C between the large negative temperature in the winter and the road surface temperature during the construction period. After the road is built, it will form common transverse shrinkage cracking due to the shrinkage of the pavement structure after the cooling temperature during winter. Therefore,

the transverse shrinkage cracking of asphalt pavement is not only related to the material properties of asphalt concrete but also to the natural environment state [33,34].

The construction period of the base layer began on 4 June 2017 (+10 °C–+21 °C). The date of the first survey was 6 April 2018 (initial melting period, −5 °C–+4 °C); the second survey was conducted on 21 October 2018 (initial freezing period, +5 °C–+17 °C); and the third survey was conducted on 17 March 2019 (late freezing period, −6 °C–+4 °C). After three consecutive investigations of transverse shrinkage cracking of asphalt pavement, the results of the mean cracking distance are shown in Figure 5.

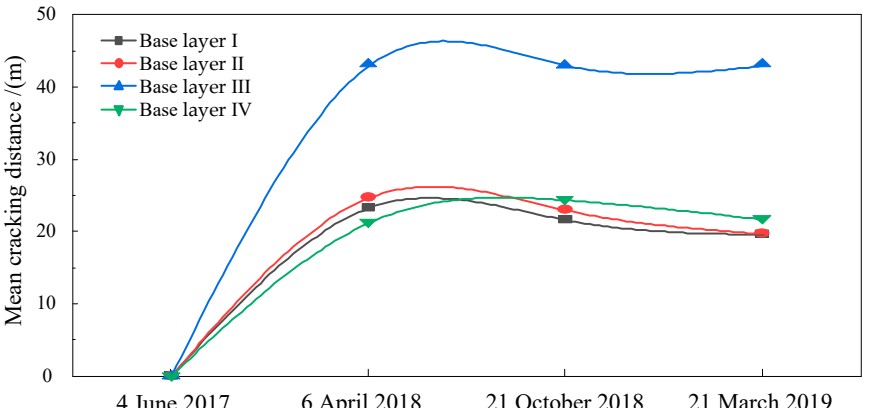

**Figure 5.** Mean cracking distance of transverse shrinkage cracking.

It can be seen from Figure 5 that after one year (6 April 2018–17 March 2019) the mean cracking distance of transverse shrinkage cracking of asphalt pavement has no obvious change. Among them, the average of the Base layer I test section was 23.1 m, and compared with the previous year little change can be observed. While the Base layer II test section remained unchanged at 43.3 m, the Base layer III test section added a half cracking, so the mean cracking distance slightly decreased. Due to the increase in the number of cracks and the development of the original cracking, the Base layer I test section was slightly reduced again, with an average of 20.8 m. In the results of the third survey, the Base layer III test section remained unchanged, while the original two half cracks of the Base layer II test section were all cracked, so the mean cracking distance was reduced to 19.8 m, which was comparable to the Base layer I test section. At the same time, the averaged cracking width of transverse shrinkage cracking of asphalt pavement was calculated as shown in Figure 6.

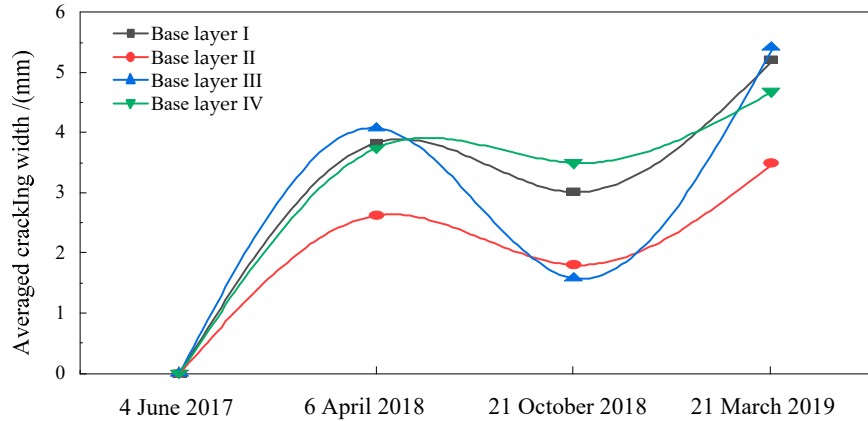

**Figure 6.** Averaged cracking width of transverse shrinkage cracking.

It can be found from Figure 6 that after half a year (6 April 2018–17 March 2019), the averaged cracking width of the Base layer I test section was 3.26 mm. Meanwhile, the Base layer II test section was 1.81 mm, and the Base layer III test section was smaller, reduced by 50% compared to the Base layer I test section. Compared with the data on 6 April

2018, it can be seen that the Base layer I test section returned from 3.80 mm to 3.26 mm, the Base layer II test section returned from 2.63 mm to 1.81 mm, and the Base layer III test section returned from 4.08 mm to 1.58 mm. It can be seen that the Base layer I test section has poor deformation recovery ability when the average temperature difference between two times was about 10 °C, and the permanent deformation (cracking width) was not recoverable due to shrinkage problems. Therefore, the fiber graded gravel has good deformation recovery ability.

The calculation results of the *CAR* of transverse shrinkage cracking of asphalt pavement are shown in Figure 7.

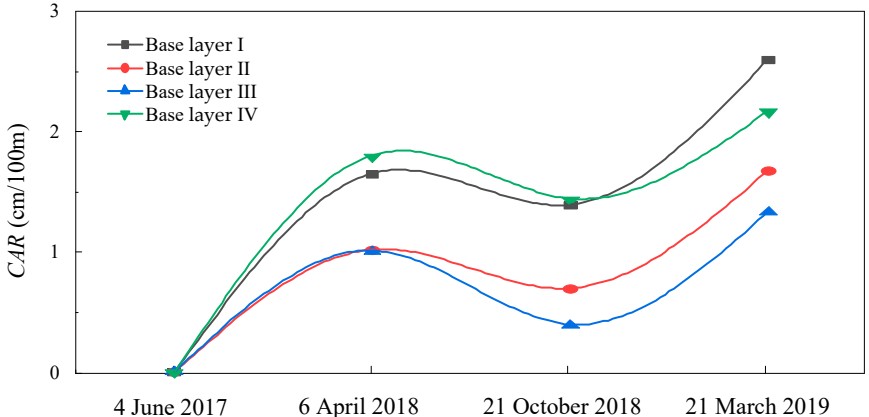

**Figure 7.** The *CAR* of transverse shrinkage cracking.

It can be seen in Figure 7 that the average *CAR* of the Base layer I test section after half a year was 1.41 cm/100 m, and the Base layer II test section has a significant advantage, which was reduced by 50% and 70% compared with the Base layer I test section, to 0.69 cm/100 m and 0.39 cm/100 m, respectively. Combined with the analysis of averaged cracking width, it can be seen that in the period of high temperature and high rainfall in summer (about four months from June to September), the *CAR* and averaged cracking width of the flexible base section (whether or not mixed with a small amount of cement) were reduced by at least 45% compared with the conventional cement gravel section, showing relatively perfect waterproof ability. From the perspective of *CAR*, the Base layer I test section has an average of 2.38 cm/100 m, while the fiber graded gravel base section (whether or not mixed with 1.25% cement) has obvious advantages, and was still nearly 40% less than the Base layer I test section, with an average of 1.50 cm/100 m.

### 4.2. Characteristics of Temperature Shrinkage Coefficient

With respect to pavement transverse shrinkage cracking, it is more beneficial to simply reverse whether the overall pavement still belongs to the category of linear temperature shrinkage according to the material temperature shrinkage coefficient formula (Formula (2)), and then judge the overall pavement dry shrinkage state as follows [35–37].

$$\alpha_L \times \Delta t \times 1000 = CAR \tag{2}$$

where, $\alpha_L$ is the linear temperature shrinkage coefficient of the material (total material of pavement structural layer); (1/°C); $\Delta t$ is the absolute value of temperature difference (average temperature of each pavement layer during construction), (°C); *CAR* is the cracking area ratio, (cm/100 m).

The calculation results of the temperature shrinkage coefficient for asphalt pavement are shown in Table 3.

**Table 3.** Characteristics of temperature shrinkage coefficient (1/ °C).

| Date | Base Layer I | Base Layer II | Base Layer III | Base Layer IV |
|---|---|---|---|---|
| 6 April 2018 | $10.3 \times 10^{-6}$ | $6.3 \times 10^{-6}$ | $6.3 \times 10^{-6}$ | $11.2 \times 10^{-6}$ |
| 21 October 2018 | $30.7 \times 10^{-6}$ | $15.3 \times 10^{-6}$ | $8.67 \times 10^{-6}$ | $31.8 \times 10^{-6}$ |
| 21 March 2019 | $15.7 \times 10^{-6}$ | $10.1 \times 10^{-6}$ | $8.1 \times 10^{-6}$ | $13.1 \times 10^{-6}$ |
| Ranges | $6.0 \times 10^{-6}$–$9.0 \times 10^{-6}$ | $6.0 \times 10^{-6}$–$9.0 \times 10^{-6}$ | $6.0 \times 10^{-6}$–$9.0 \times 10^{-6}$ | $6.0 \times 10^{-6}$–$9.0 \times 10^{-6}$ |

As shown in Table 3, following a complete freezing period, the temperature shrinkage coefficient of the Base layer III test section was at the lower limit of the normal value, indicating excellent temperature shrinkage performance. However, the temperature shrinkage coefficient value of the Base layer I test section slightly exceeded the upper limit of the normal value, which led to poor temperature shrinkage performance. Despite more than a year of operation, the Base layer III test section still exhibited sound temperature expansion and contraction performance. On the other hand, the Base layer II test section exceeded the upper limit of the normal value, indicating that cracking caused by a dry shrinkage component had occurred, with slightly poor temperature shrinkage and deformation recovery ability. Furthermore, the temperature shrinkage coefficient value of the Base layer I test section was seriously beyond the limit with poor thermal expansion and cold contraction performance. After nearly two years of operation, the Base layer III test section continues to demonstrate good performance in terms of temperature shrinkage and deformation recovery, while the Base layer II test section slightly exceeds the overall linear expansion coefficient of the normal pavement structure layer, indicating obvious dry shrinkage component, and slightly weak ability of temperature shrinkage and deformation recovery.

In summary, different asphalt pavement base layers have different influences on transverse shrinkage cracking characteristics, among which, asphalt pavement with the Base layer III test section has a relatively good performance. Compared with the traditional cracking indexes, *CAR* could comprehensively analyze the transverse shrinkage cracking detection data, which had the advantages of being intuitive, comprehensive, scientific, reasonable, and practical. The proportion of *CAR* in the Base layer II test section was slightly different, but it was still close to and obviously superior to that of the Base layer I test section. For a flexible base layer, proper thickening of the pavement surface could only delay the occurrence and development of transverse shrinkage cracking. From the perspective of temperature shrinkage coefficient, the temperature shrinkage coefficient of the Base layer III test section was at the upper limit of the normal value and showed better temperature shrinkage and deformation recovery performance. Additionally, the Base layer II test section had exceeded the overall linear expansion coefficient of the normal pavement structure layer (Base layer I), the cracking was obviously caused by the dry shrinkage component, and the deformation recovery ability was slightly poor. This is because the shrinkage component caused by the cracking is large, the recovery ability from cracking at high temperatures is poor, and the permanent cracking (unrecoverable deformation) is obvious.

## 5. Conclusions

In the study, combined with the engineering project and through the actual investigation method, it is concluded that the transverse shrinkage cracking characteristics are affected by the asphalt pavement base layers, and the main conclusions are as follows:

(1) The study defined the cracking area ratio (*CAR*), which, for transverse shrinkage cracking, has a direct significance in that it can reflect the overall longitudinal shrinkage of the section.

(2) The overall shrinkage of conventional cement sand gravel mixed with gravel (Base layer I) is larger, the anti-shrinkage cracking pavement performance of fiber graded gravel (Base layer III) is better, and the overall anti-shrinkage cracking performance of the asphalt pavement is not affected by 1.25% cement.

(3) For fiber graded gravel with or without cement, the mean cracking distance and averaged cracking width of transverse shrinkage cracking vary greatly, but the *CAR* is similar.

(4) Fiber graded gravel base can effectively relieve the load stress of asphalt pavement and prevent the upward reflection cracking in the middle and lower base.

(5) Asphalt pavement with fiber graded gravel (Base layer III) shows good temperature shrinkage performance in the whole detection cycle, followed by Base layer II, Base layer IV, and Base layer I.

**Author Contributions:** Conceptualization, T.Y. and M.X.; methodology, M.X.; validation, M.X., H.Y. and Z.Z.; formal analysis, H.Y.; investigation, Z.Z.; writing—original draft preparation, T.Y.; supervision, M.X. All authors have read and agreed to the published version of the manuscript.

**Funding:** This research was funded by Science and Technology Projects of Heilongjiang Provincial Department of Transportation, Application of High-Performance Graded Gravel Base to Improve the Overall Performance of Asphalt Pavement in Cold Area, grant number [HJK2016B016]; Research on Countermeasures of Asphalt Pavement Against Longitudinal Cracks Based on Water Temperature Characteristics of Highway Subgrade in Cold Region, grant number [HJK2019B019].

**Institutional Review Board Statement:** Not applicable.

**Informed Consent Statement:** Not applicable.

**Data Availability Statement:** Not applicable.

**Acknowledgments:** The authors gratefully appreciate the support from the projects of Science and Technology Project of the Transportation Department of Heilongjiang Province, Research on the application of coal gangue burning slag in the base and depth of highway pavement (No: HJK2019B003).

**Conflicts of Interest:** The authors declare no conflict of interest.

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
