# Peer review of "Effect of Asphalt Pavement Base Layers on Transverse Shrinkage Cracking Characteristics"

_sustainability, doi:10.3390/su15097178_

Round 1

Reviewer 1 Report

Before making any recommendations for a interesting scientific article “Effect of base course types on characteristics of transverse temperature shrinkage crack for asphalt pavement”, I would like to present the following statements on the topic. Based on my long-term research and transfer profile in the field of holistic perception of the issues of civil engineering structures especially pavements, I consider the evaluated scientific article to be topical and fully convergent with the following author's long term research and educational premise.  Pavements should be designed, built, managed, maintained, recycled (decomposed) at a reasonable price, in reasonable quality, respecting the relevant requirements of users, residents and sustainable development, including saving non-renewable resources and circular economy. An integral part of the presented premise is undoubtedly the prevention of temperature shrinkage crack of asphalt pavements.

In order to improve the contribution enabling its publication in a renowned scientific journal Sustainability, I would like to recommend incorporation of the following mandatory and facultative requirements respectively a credible justification for their non-implentation into revision of reviewed article.

Mandatory requirements:

LNSA 9-26…Abstract…  I would allow myself to take the following position regarding the abstract. In general, I expect from a high-quality scientific abstract a brief scientific summary of the solved problem and from it resulted hypotheses, an explicit determination of scientific goals and corresponding methodology. The sentence " To investigate the effect of base course types on the transverse temperature shrinkage crack for asphalt pavement, the study collected the transverse temperature shrinkage crack data of different base course types test section and evaluated the characteristics of base course on the transverse temperature shrinkage crack"  it is necessary to divide into at least 2 sentences and rewrite the texts in order to make them more scientifically expressive to the subject of the article.

LNSA 47-49… Tian [6] found that the tensile stress of the lower base course was greater than that of the upper base course, and the fatigue life of the pavement structure depended on the characteristics of the base course… This sentence does not have general validity without further specification. It is necessary to briefly provide additional information on the structural composition of the roadway, traffic, climatic and mechanical characteristics of the structural layers. Tensile stress is significantly influenced by air temperature, more details can be found, for example, in "Evaluation of the Effect of Average Annual Temperatures in Slovakia between 1971 and 2020 on Stresses in Rigid Pavements. Land, 11(6), 764".

LNSA 90...Figure 1 Geographic location of the project [14]... it would be appropriate to territorially identify Heilongjiang Province within the whole of China or the corresponding Asian subcontinent.

LNSA 132… Figure 3 Transverse temperature shrinkage crack of asphalt pavement... it is necessary to increase the graphic quality of the figure (increase the graphic resolutions of the photos).

LNSA 141...3. Analysis of transverse shrinkage crack characteristics for asphalt pavement...the chapter title  to be moved to the next page.

LNSA 213… Figure 4 Mean crack distance D and average crack width Bf for transverse temperature shrinkage

cracks (a) Mean crack distance D; (b) Average crack width Bf....it is necessary to significantly increase the graphic quality of the figure and add numerical data to the column representations. This requirement also applies to figure 5.

LNSA 272...Figure 6 Mean crack distance of transverse temperature shrinkage crack..it is necessary to increase the graphic quality of the figure and consider the appropriateness of the type of graph used. This requirement also applies to figure 7 and 8.

LNSA 403-426… Conclusions ... conclusions need to be headed a comprehensive manner with emphasis to the wider scientific context of the article. It is necessary to avoid simplex text constatations  and strengthen the cognitive aspect of the conclusions. The findings of the authors need to be confronted with the credible works of renovated foreign authors.  Conclusions should not end the numbered paragraph of the text, they should predict or at least anticipate credible continuations of research in the subject area.

Facultative recommendations:

LNSA 2-3...Effect of base course types on characteristics of transverse temperature shrinkage crack for asphalt pavement... I would like to recommend to the authors a slight modification of the title of reviewed scientific article. Personally, I would use, for example, the following title: Effect of asphalt pavement base layers to transverse shrinkage crack characteristics (please take it only as an inspiration).

LNSA 86… average annual temperature is between 1.3 ℃ and 4.0 ℃… average annual temperature primarily depends on the altitude of the pavements and therefore it would be appropriate to state this data for the stated temperatures. More details can be found, for example, Sustainable Adaptive Cycle Pavements Using Composite Foam Concrete at High Altitudes in Central Europe. Sustainability, 14(15), 9034.

LNSA 154-155...Where, Pa is proportion of fracture area, (cm/100m); Bs is the total width of cracks in the section, (mm); L is the total length of the section is counted,…Why are different multiples of length units (mm, cm, m) used?

LNSA 175… Table 2 Calculation results of transverse temperature shrinkage crack on asphalt pavement... I recommend reformatting the table (font sizes, ..) so that the table fits on 1 page and it is necessary to insert an empty line under the table.

LNSA 343... Table 3 Characteristics of temperature shrinkage coefficient... it is necessary to insert an empty line above the name of the table and use a more suitable format for writing numerical data

LNSA 444-500… References… I personally consider 25 references to be insufficient. For this type of scientific article, in my view, the number of 35 to 40 is suitable, I also recommend considering the implementation of the works of European authors.

I consider the scientific potential of the reviewed scientific article to be considerable and, despite the mentioned comments, I recommend its revision. I would like to  repeated recommendations on  significant increase in quality of figures, expansion of related references, including European sources, and  comparison of the author's findings with credible foreign scientific works.

Reviewer 2 Report

This research studied the effect of the base course on the transverse temperature shrinkage crack for the asphalt pavement, where the data collected was based on actual survey results. In general, the results have been well presented and discussed. The following comments need to be addressed by the authors to further improve the manuscript:

·        In the abstract, the main problem of statement and/or the gap of the presented study need to be clearly highlighted.

·        Also, the Introduction needs to clearly discuss the technical problem of statement of the suggested study or could be revise the existing sentence for better understanding by the reader.

·        As possible, Improve the text’s quality of Figures 2,4, and 5.

·        In the conclusions, it is advised to clearly highlight the limitations of this study. 

Reviewer 3 Report

- The abstract is too long. Make it simple and firm. The section should be included the short objective, problem statement, major finding and significant study. Please also include the analytical data.  

- The introduction is well written. However, it is required the correlation between paragraph.  

- The discussion and analysis of the result have been clearly explained but the authors lacks in the correlation between subsection  

- The conclusion should be revise with more details on the main finding.  

- The references are outdated. Please replace to the updated version ( 5 years before current year)

Reviewer 4 Report

The author invesitageted the effects of base course types on the transverse temperature shrinkage cracks for asphalt pavement. The finding is interesting but the reviewer believes that the paper does not fit in the scope of paper.

The authors should explain how the content fit the scope of the sustanability.

Author Response

Dear Reviewer:

Thank you very much for your comments and suggestions. After careful reading of the aims and scope of the special issue, we found that our research content fits within the scope of the special issue.

Firstly, the special issue mainly focuses on "new theory, methodology, and technology satisfying the sustainability, durability, safety, and environmental protection of public facilities in cold regions of the world." Our research subject is road engineering facilities in northeast China, which belong to public facilities in cold regions of the world. At the same time, the research of low temperature cracks on asphalt pavement belongs to Topic 5 (Engineering in cold regions) and Topic 7 (Construction technology in cold regions and artificial freezing).

Secondly, according to relevant research and engineering investigation, the service life of asphalt pavement is directly affected by low temperature cracks (especially transverse shrinkage cracks). The appearance of cracks makes the structure layer of asphalt pavement easy to be damaged by precipitation or icing, thus accelerating the failure of asphalt pavement performance. It will increase the cost of construction and maintenance of road engineering, which is not conducive to the sustainable development of road engineering. Therefore, asphalt pavement with fewer transverse cracks can provide a longer service life, that is, a sustainability value.

Finally, we will modify the article in accordance with the Reviewer's comments. Add the impact and significance of the research on the sustainable development of asphalt pavement by crack disease (transverse crack disease), so as to be more consistent with the scope of the journal.

We tried our best to improve the manuscript and made some changes in the manuscript. We appreciate for Reviewer' warm work earnestly, and hope that the correction will meet with approval.

Thank you very much for your review.

Reviewer 5 Report

1. The aim of the presented paper is to evaluate the influence of base course type on transverse shrinkage crack of asphalt pavements.

2. The style and clarity of the article's text need significant revision and improvement. In its current form, the article reads difficult and is incomprehensible in places. Examples of ambiguities are shown below.

L. 73-74: However, the effect of base course types on the characteristics of transverse temperature shrinkage crack in asphalt pavement has not been explored and explored.

L. 100-105: Combined with the structural layer information of the asphalt pavement in the test section (as shown in Table 2), it can be seen that the transverse crack corresponding to the boundary between K2+425 semi-rigid base and flexible base, the transverse crack corresponding to the boundary between K2+525 fiber graded gravel base and whether a small amount of cement is added, and the transverse crack corresponding to the boundary between the second and third layers of K2+575 pavement layer respectively, and the error is less than 5m.

L. 108-109: In the calculation and analysis, we have taken into full consideration the problems of section division and section division.

L.111-113: Asphalt pavement crack detection is one of the important components for road engineering detection, and its detection results are directly related to the evaluation process of road technical status [18].

3. Table 1 shows the fracture parameters of road structures. In column 2, the description of the same substructure in each of the comparison rows is switched. What is the point of comparing data on the same case and what is the result? According to the reviewer, this information should be explained in the text, and the repeated description of the layers in column 2 is unnecessary and confusing.

4. Equation 1 presents the proportion of fracture area Pa, (cm/100m). While we are discussing the area, and the units are cm/100 m, why is the equation presenting the width of cracks (mm x 10) divided by the length counted in meters. It is confusing and needs explanation, Pa is not the area, and the resultant units according to the formula shown are (cm/m).

5. In the reviewer's opinion, the abstract should not indicate strict numerical results of the research.

6. The research and results seem to address a common and described problem. in the opinion of the reviewer, their new contribution to the field of knowledge addressed should be clearly proven. The discussion of the research results should also be completed and differences and novelties with respect to the current literature on the subject should be pointed out.

7. How does the research relate to the problem of sustainability in construction?

Yours Sincerely,

Reviewer.

Round 2

Reviewer 1 Report

Based on the detailed inspection of  incorporation of the changes recommended by me, I allow myself to rate the assessed second version of the scientific paper as follows. Reviewed contribution “Effect of asphalt pavement base layers to transverse shrinkage crack characteristics”, original title: “ Effect of base course types on characteristics of transverse temperature shrinkage crack for asphalt pavement”.  Based on my holistic  experience in the assessed issue and subsequent deepening of my knowledge, I am pleased that the submitted 2nd version of the article meets all my essential requirements for a quality scientific article.

In conclusion, I would like to sincerely congratulate the authors on a very good scientific article and thank the publisher for the opportunity to expand my scientific knowledge in the following field. Creation and influence of transverse shrinkage cracks, as the main factor leading to the degradation of performance for asphalt pavements.

Author Response

Dear Reviewer:

We tried our best to improve the language expression of the manuscript. With the help of colleagues majoring in English, we made some modifications to the manuscript. These changes will not influence the content and framework of the paper. We appreciate for Reviewer' warm work earnestly, and hope that the correction will meet with approval.

Thank you very much for your comments and suggestions.

Reviewer 4 Report

The paper can be accepted

Author Response

(The authors gave the same response as above.)

Reviewer 5 Report

Dear Authors

In my opinion, taking into account the journal to which the article is submitted should more clearly indicate the problem of sustainability in construction in relation to the subject of the work covered.

Author Response

(The authors gave the same response as above.)
